# OSVI-WM: One-Shot Visual Imitation for Unseen Tasks using World-Model-Guided Trajectory Generation

**Raktim Gautam Goswami[1]  Prashanth Krishnamurthy[1]  Yann LeCun[2,3]  Farshad Khorrami[1]**

[1]New York University Tandon School of Engineering
[2]New York University Courant Institute of Mathematical Sciences   [3]Meta-FAIR

## Abstract

Visual imitation learning enables robotic agents to acquire skills by observing expert demonstration videos. In the one-shot setting, the agent generates a policy after observing a single expert demonstration without additional fine-tuning. Existing approaches typically train and evaluate on the same set of tasks, varying only object configurations, and struggle to generalize to unseen tasks with different semantic or structural requirements. While some recent methods attempt to address this, they exhibit low success rates on *hard* test tasks that, despite being visually similar to some training tasks, differ in context and require distinct responses. Additionally, most existing methods lack an explicit model of environment dynamics, limiting their ability to reason about future states. To address these limitations, we propose a novel framework for one-shot visual imitation learning via world-model-guided trajectory generation. Given an expert demonstration video and the agent's initial observation, our method leverages a learned world model to predict a sequence of latent states and actions. This latent trajectory is then decoded into physical waypoints that guide the agent's execution. Our method is evaluated on two simulated benchmarks and three real-world robotic platforms, where it consistently outperforms prior approaches, with over 30% improvement in some cases. The code is available at https://github.com/raktimgg/osvi-wm.

## 1   Introduction

*Imitation is the sincerest of flattery.*

— Charles Caleb Colton[1]

Intelligent beings like humans are capable of learning a wide range of skills by observing and imitating others [50]. Even toddlers, with a basic understanding of the world's dynamics, solve complex tasks by watching *expert demonstrations*. Inspired by this natural learning process, principles have been applied in robotics [13, 26, 35, 62] where expert demonstrations, typically collected through human teleoperation, are used to teach agents to perform tasks through imitation. Imitation learning finds uses across diverse domains, including medical robotics, collaborative robotics, and industrial automation.

In a successful imitation, the agent discovers the skill demonstrated by the expert, adapts it to its own embodiment, and executes it within its environment [58]. In the one-shot visual imitation (OSVI) setting, the agent must derive a policy from a single expert demonstration video, without any additional training [12]. Notably, the expert and agent may differ in embodiment [10]; in fact, both prior work [51, 43, 37] and our approach have utilized human demonstration videos as input. Existing methods, however, often rely on the strong assumption that the training and testing tasks are nearly identical, typically differing only in object locations. As a result, generalization

---

[1]*Lacon: Or Many Things in Few Words*, 1820.

39th Conference on Neural Information Processing Systems (NeurIPS 2025).

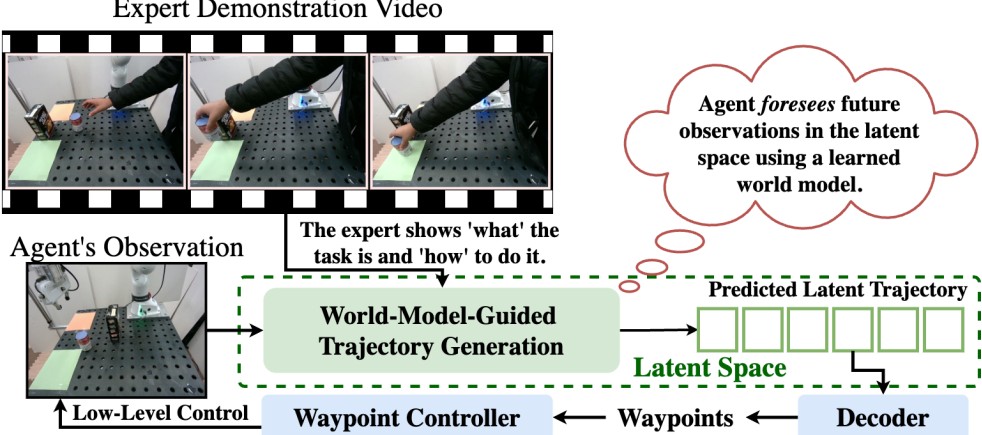

Figure 1: OSVI-WM infers the task from the expert demonstration and, along with the agent's observation "foresees" future latent states using a world-model-guided trajectory generation module. The predicted trajectory is decoded into physical waypoints for control.

to semantically or structurally different tasks remains limited. MOSAIC [34] and AWDA [5] were among the first to investigate generalization to unseen tasks. In particular, AWDA [5] demonstrated that the performance of standard methods like [10, 60] degrades severely when evaluated on tasks that differ even marginally from those seen during training. Although AWDA outperforms earlier methods [10, 60], it exhibits low success rates on test tasks that are visually similar but contextually different, such as opening vs. closing a sliding window. Moreover, none of these approaches incorporates an explicit model of the environment's dynamics, limiting their ability to reason about long-term consequences or plan into the future.

Mathematically modeling the environment using raw camera images is challenging due to the presence of diverse objects and their complex interactions. To address this, neural networks have been used to learn world models that predict how the environment will evolve in response to the agent's actions. These models support applications in reinforcement learning [48, 21, 28, 39, 6, 23, 56] and behavior cloning [8, 37]. Inspired by such advances, we propose *OSVI-WM* (Fig. 1), a framework for one-shot visual imitation for unseen tasks using world-model-guided trajectory generation. Given an expert demonstration video and the agent's initial observation, OSVI-WM encodes each image into a shared latent space representing the environment state for the respective image. These embeddings are recursively processed through learnable action and world models to predict a trajectory of future latent states, allowing the agent to plan into the future. Predicting actions in the latent space also helps mitigate the multi-modality of real-world action distributions. The predicted latent trajectory is then decoded into physical waypoints [5] using pooling and Multi-Layer Perceptron (MLP) layers.

The predicted waypoints guide the agent's execution and provide supervision for end-to-end training. To enhance the world model's ability to capture environment dynamics, an auxiliary loss is applied on the predicted latent trajectory during training. Further, OSVI-WM supports re-planning: if the agent doesn't accurately complete the task by following the predicted waypoints, it takes a new observation and plans a fresh set of waypoints based on its current state. Notably, unlike methods that rely on large-scale pretraining or extensive demonstration data [37, 3, 55], OSVI-WM is trained directly within the task domain. In summary, our main contributions are:

- An efficient end-to-end imitation learning architecture trained solely on in-domain data, without requiring large-scale pretraining.

- A novel world-model-guided trajectory generation module tailored for OSVI on unseen tasks.

- Robustness enhancement at test time by using a waypoint controller with re-planning.

- Extensive experiments in both simulated and real-world settings, demonstrating that OSVI-WM outperforms existing methods on unseen tasks.

## 2 Related Works

**One-Shot Imitation Learning**: Imitation learning, or learning from demonstrations, is broadly categorized into inverse reinforcement learning [1, 40] and behavior cloning [46, 31, 7, 22, 42]. While the field has been extensively studied for years, a comprehensive review is beyond the scope of this paper; we, therefore, refer readers to existing surveys [13, 35, 62]. Our method falls under behavior cloning, specifically in the one-shot visual imitation (OSVI) setting. In standard behavior cloning, expert demonstrations and corresponding expert actions are available to train the agent. In contrast, OSVI [12] provides only a video of the expert performing the task, without access to the underlying actions. One of the earliest works on OSVI [12] introduced the problem setting and proposed a soft-attention-based learning framework for block stacking. Since then, a number of methods [15, 10, 4, 63, 27] have improved performance in this setting. While ego-centric setups (e.g., DINOBot [11]) have been explored, we focus on methods using external camera views. One such work, T-OSVI [10], uses transformers [49] to model long-range dependencies in demonstrations, and like many other works [29, 51, 60], it supports expert-agent embodiment mismatch.

While these approaches have shown promising results, they are typically evaluated on the same tasks that were used for training with minor variations, such as changes in object positions or quantities. MOSAIC [34] proposed a more challenging setup with completely unseen test tasks, building on which, AWDA [5] demonstrated that existing methods like T-OSVI [10] and DAML [60] perform poorly in such settings. AWDA improved generalization through attributed waypoints, demonstration augmentation, and image mixup, but achieved low success rates on *hard* tasks requiring different responses despite visual similarity. While IMOP [63] used point clouds to aid generalization, we focus solely on monocular RGB images, which are more accessible and require no specialized hardware. Notably, none of these approaches models environment dynamics, limiting their ability to reason about future states. OSVI-WM addresses this gap through a world-model-guided trajectory generation module that enables future-aware planning and improved generalization to unseen tasks.

**World Model in Robotics**: Intelligent beings like animals are believed to possess internal models of the world that support control and planning for task execution [52, 53, 16, 38]. Inspired by this, researchers have increasingly explored the use of world models in robotics [45]. In reinforcement learning (RL), several works [17, 28, 19–21, 18, 39, 56, 6, 44, 14] have focused on learning world models from agent rollouts. These models are then used as environments for RL policy training, often also predicting rewards based on the agent's actions. Most prior works train world models directly in the pixel space, i.e., conditioned on the current image and action, the model predicts the next-frame image, often also integrated with diffusion-based frameworks. These world models, however, requires modeling low-level pixel details that are often irrelevant for downstream tasks [2]. To address this, recent methods [24, 23, 36, 30] learn world models in a latent embedding space, allowing for more efficient and abstract representations of the environment.

Beyond RL, world models have been explored for broader robotic learning tasks. For instance, SWIM [37] pre-trains a world model from human videos and fine-tunes it on robot data in an unsupervised manner. GR1 [54] and [59] propose pre-training strategies to improve downstream robotic motor control. Most of these approaches, however, rely on large-scale internet data or massive pretraining datasets, leading to high computational demands. DynaMo [8], in contrast, proposed in-domain latent dynamics pretraining for the encoder, which was used to train downstream policies. Similarly, DINO-WM [64] used the pre-trained DINOv2 [41] model (with frozen weights) to build in-context world models and applied model predictive control based optimization for task execution. Inspired by these advances, our proposed framework, OSVI-WM, learns a world model entirely from in-domain data. It predicts latent trajectories based on an expert demonstration and the agent's current observation. Unlike prior work, OSVI-WM, including its world model, is trained end-to-end, without any pretraining, simplifying the pipeline and reducing computational complexity.

## 3 Our Method

**Problem Formulation**: We formulate the problem following prior work in OSVI [10, 34, 5]. Let $\mathcal{T} = \{\mathcal{T}_1, \ldots, \mathcal{T}_K\}$ denote a set of tasks, partitioned into disjoint training ($\mathcal{T}_{train}$) and test ($\mathcal{T}_{test}$) sets. In contrast to [10] and consistent with [5], our setup uses different training and test tasks, rather than minor variations of the same task. Each training task consists of multiple sequences of expert and agent trajectories. The expert trajectory comprises demonstration video frames $\{E_1, \ldots, E_N\}$, while

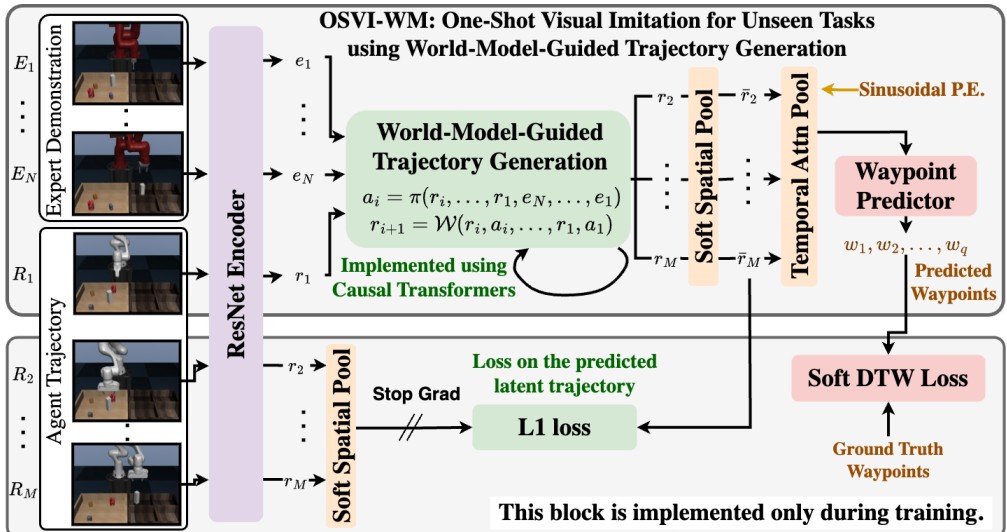

Figure 2: **OSVI-WM**: The expert demonstration frames $E_1, \ldots, E_N$ and the agent's current observation $R_1$ are encoded into a latent space using a ResNet encoder. A world-model-guided trajectory generation module predicts future latent states $r_2, \ldots, r_M$, which are decoded into physical waypoints for control. During training, supervision is applied on the predicted waypoints and latent states.

the agent trajectories are represented by observation-state pairs $(\mathcal{R}, \mathcal{S})$ where $\mathcal{R} = \{R_1, \ldots, R_M\}$ denotes the observation frames and $\mathcal{S}$ denotes the corresponding agent states. Within each task, all expert and agent trajectories correspond to variations (e.g., different object configurations) of the same high-level task. The model, trained on $\mathcal{T}_{train}$, is evaluated on $\mathcal{T}_{test}$, which contains only expert demonstration videos (no agent rollouts). The performance is measured by recording the success rate of the agent in completing these test tasks. The key challenges in this setup are: (a) test tasks $\mathcal{T}_{test}$ differ from the training tasks $\mathcal{T}_{train}$; (b) the expert and the agent may have different embodiments; (c) there is no direct alignment between expert and agent states or actions; and (d) to avoid reliance on costly large-scale data, the method should be trained in-domain.

**Method Overview**: Given expert demonstration video frames $E_1, \ldots, E_N$ and the agent's initial observation $R_1$, OSVI-WM (Fig. 2) encodes them using a shared ResNet-18 [25] encoder to obtain latent states for the expert and agent. These are passed to the world-model-guided trajectory generation module (Sec. 3.1) to predict a trajectory of future latent states, which is pooled, first spatially, then temporally, into a compact representation. Using this as input, an MLP-based waypoint predictor generates physical waypoints (Sec. 3.2). Loss functions are applied to the predicted waypoints and the latent states predicted by the world model (Sec. 3.3).

## 3.1 World-Model-Guided Trajectory Generation

The expert demonstration and the agent's initial observation frames are each processed through a ResNet-18 [25] encoder, producing feature maps of shape $(F, H, W)$ before the encoder's global pooling layer, where $F = 512$ and $(H, W)$ depend on the input resolution. These features are flattened to produce latent states for the expert frames $(e_1, \ldots, e_N)$ and the agent's observation $(r_1)$, each frame with shape $(F \times H \times W)$. A latent state is a compact, abstract representation of the environment that encodes key features such as the configuration of objects and dynamic elements. A latent action similarly represents an abstract transformation that drives state transitions. Operating in this latent space further helps mitigate the multi-modality of real-world action distributions. The

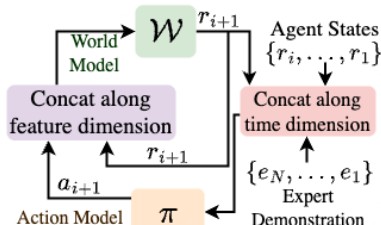

Figure 3: **WM-Guided Trajectory Generation**: Starting with the agent's initial observation and the expert demonstration, future states are recursively predicted using action and world models.

world-model-guided trajectory generation module consists of an action model $\pi$ and world model $\mathcal{W}$. At timestep $i$, $\pi$ predicts latent action $a_i$ using the agent's current and past latent states along with the

expert demonstration's latent states. This action is used by $\mathcal{W}$ to predict the agent's next latent state:

$$a_i = \pi(r_i, \ldots, r_1, e_N, \ldots, e_1), \quad r_{i+1} = \mathcal{W}(r_i, a_i, \ldots, r_1, a_1); \quad i = \{1, \ldots, M-1\}. \quad (1)$$

This multi-step recursive process (Fig. 3) generates all future latent states of the agent $(r_2, \ldots, r_M)$, forming the generated trajectory. The world model (WM) essentially learns the environment's dynamics to predict the next latent state based on the current latent state and latent action.

Within the action model $\pi$, at each step $i$, the expert states $\{e_N, \ldots, e_1\}$ and agent states $\{r_i, \ldots, r_1\}$ are concatenated along the temporal dimension, resulting in shape $(N+i, F \times H \times W)$. This is processed through causal transformer blocks, and the output corresponding to the final temporal index is used as the latent action $a_i$. For the world model $\mathcal{W}$, the latent actions $\{a_i, \ldots, a_1\}$ are concatenated with the corresponding agent states $\{r_i, \ldots, r_1\}$ along the feature dimension to obtain shape $(i, 2F \times H \times W)$. This sequence is processed through a causal transformer, and the output at the last time step is projected to shape $(F \times H \times W)$ to produce the next state $r_{i+1}$.

## 3.2 Predicting Waypoints

**Spatial and Temporal Pooling**: The predicted trajectory of future states $(r_2, \ldots, r_M)$ is reshaped back to shape $(F, H, W)$ at each timestep to retain spatial structure. A spatial pooling module [10] then applies a 2D softmax across each feature map and computes an expected 2D coordinate over a $[-1, 1]$ grid for each feature dimension. These coordinates are concatenated, producing a spatially pooled output of shape $2F$ for each of the $M-1$ timesteps $(\bar{r}_2, \ldots, \bar{r}_M)$. Sinusoidal positional embeddings are then added along the temporal dimension, and an attention pooling module with a learned query summarizes the sequence into a single $2F$-dimensional vector.

**Attributed Waypoint Prediction**: The pooled vector is passed through a two-layer MLP to predict 5 attributed waypoints, each of dimension 4. Following [5], each waypoint encodes the end-effector's 3D position in the camera's coordinate frame and a binary gripper state (open/closed). These positions are transformed into the robot's coordinate frame using a known camera-to-robot transformation. A waypoint controller, using inverse kinematics of the robot, computes low-level control commands for execution. To ensure accurate grasping, especially in the final centimeters before contact, we follow [5] and use an end-effector-mounted depth camera to correct residual pose errors.

**Waypoint Re-Planning**: To improve robustness during execution, OSVI-WM supports waypoint re-planning, which activates if the agent fails to complete the task using the predicted waypoints. In simulation, success or failure is automatically determined from environment configurations. Although we do not use re-planning on real-world benchmarks in this work, success (or completion) signals could be obtained via learned reward/value models [24] or vision–language-based completion detectors [32, 57]. In OSVI-WM, once all waypoints are executed, the agent captures a new observation, which, combined with the original expert demonstration, generates updated waypoints. Notably OSVI-WM outperforms existing baselines even without re-planning (Fig. 8).

## 3.3 OSVI-WM Training

**Loss Functions**: We jointly optimize the world model and the waypoint predictions. To supervise the world model, ground-truth agent observations $R_2, \ldots, R_M$ from the training set are encoded into latent states using our encoder, followed by the spatial pooling layer to produce $\hat{r}_2, \ldots, \hat{r}_M$. An $L_1$ loss is applied between these and the predicted latent states after spatial pooling $(\bar{r}_2, \ldots, \bar{r}_M)$: $\mathcal{L}_{wm} = \frac{1}{M-1} \sum_{i=2}^{M} \|\bar{r}_i - \hat{r}_i\|_1$. To prevent model collapse and ensure stable training, we apply a stop-gradient to $\hat{r}_2, \ldots, \hat{r}_M$, preventing backpropagation through the ground-truth latent states. For waypoint supervision, we interpolate the ground-truth agent trajectory into a denser sequence and use the differentiable soft dynamic time-warping loss [9] ($\mathcal{L}_{sdtw}$). Although only 5 waypoints are used at inference, we follow [5] and predict multiple sets (1 to 5 waypoints) during training, applying separate losses for each set to improve supervision across different trajectory granularities. The overall training objective is $\mathcal{L} = \mathcal{L}_{sdtw} + \alpha(\tau)\mathcal{L}_{wm}$ where $\alpha(\tau)$ is a scheduling factor that balances the two loss terms based on the training iteration $\tau$. For the Meta-World [61] benchmark experiments in Sec. 4, $\alpha(\tau)$ is initialized to 1 and decays exponentially to 0.05 by the end of training. For Human-Franka-PP, $\alpha(\tau) = 10$; for all other datasets, we keep $\alpha(\tau) = 1$.

**Training Settings**: Each expert demonstration is downsampled to $N = 10$ frames, and each agent trajectory to $M = 6$ frames. Following prior work [10, 5], we pair every expert demonstration

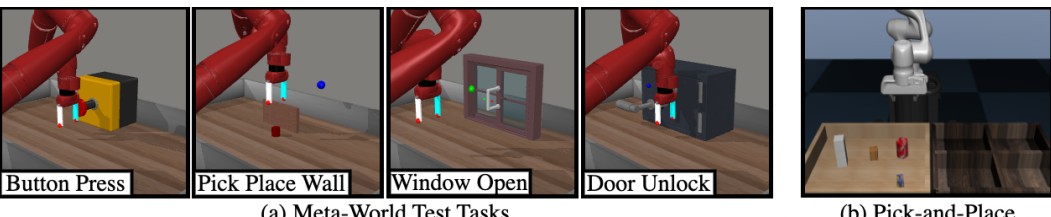

(a) Meta-World Test Tasks      (b) Pick-and-Place

Figure 4: **Simulation Environments**: Test tasks are different from the ones used for training. Additionally, Pick-and-Place uses different embodiments for expert and agent.

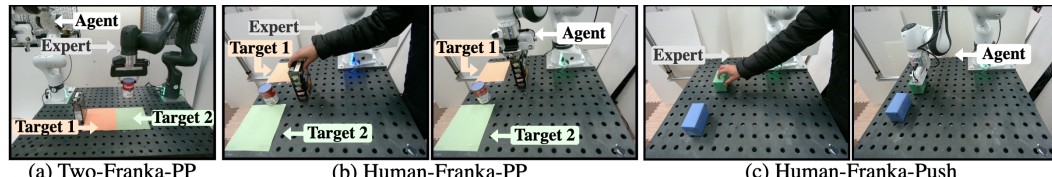

(a) Two-Franka-PP     (b) Human-Franka-PP     (c) Human-Franka-Push

Figure 5: **Real-World Environments**: (a) Pick-and-place setup with expert (gray) and agent (white) Franka arms mounted at different locations. (b) Similar setup with a human expert and Franka agent. (c) Box push setup with human expert and Franka agent.

with every agent trajectory within the same task, resulting in a combinatorial dataset expansion. We train for 10 epochs in simulation and 100 in real-world settings using AdamW [33] optimizer with a one-cycle learning rate scheduler [47] (start: 0.0002, end: $2 \times 10^{-7}$), and a batch size of 128. We apply asymmetric demonstration image mixup [5] regularization to help the model focus on task-relevant cues rather than video semantics. Although training runs for 10 epochs, we use early stopping for Meta-World to avoid overfitting and report results using the overall best-performing checkpoint. For simulation runs, we used an RTX A4000 GPU with 128 GB RAM and Intel i9 CPU, and for real-world, we used an RTX 2080 Ti, 64 GB RAM, and Intel i7 CPU.

## 4 Experiments

Our experiments are designed to address the following key questions: (a) How well does OSVI-WM generalize to novel, unseen tasks? (b) Can OSVI-WM be used for real-world robotic tasks? (c) What is the impact of the world-model-guided trajectory generation module on overall performance? (d) Which components of OSVI-WM are most critical to its success?

### 4.1 Evaluation Settings

**Environments**: We evaluate OSVI-WM on simulation (Fig. 4) and real-world benchmarks (Fig. 5):

(a) **Meta-World** [61]: A simulation benchmark of 50 manipulation tasks, split into 46 for training and 4 for testing following [5]. Test tasks are different from train tasks and are divided into 'easy' tasks (Button-Press-V2, Pick-Place-Wall-V2), which differ from training tasks due to the presence/absence of distractor, and 'hard' tasks Window-Open-V2, Door-Unlock-V2, which, despite being visually similar to Window-Close-V2 and Door-Lock-V2 from the training set, respectively, differ in context and require distinct responses.

(b) **Pick-and-Place** [10]: A simulated benchmark involving 4 objects and 4 targets, yielding 16 tasks. Following [5], we use 14 tasks for training and 2 for testing. This benchmark features different embodiments: the expert uses a Sawyer arm, and the agent uses a Franka arm.

(c) **Two-Franka-PP** (Fig. 5a): A real-world setup with two Franka arms mounted at different tabletop locations, with the gray arm as the expert and the white arm as the agent. The tasks are similar to Pick-and-Place above, with two objects and two targets (4 tasks total). We use 3 tasks for training and 1 for testing. The difference in mounting location between the expert and agent introduces additional imitation complexity.

(d) **Human-Franka-PP** (Fig. 5b): Similar to Two-Franka-Env, but the expert is a human arm and the agent is a Franka robot. This greater embodiment mismatch increases task difficulty. As before, we use 3 tasks for training and 1 for testing.

Table 1: Success rates (in %) comparison on the Meta-World [61] and Pick-and-Place [10] simulation benchmarks. Best results are highlighted. We also report if a method uses additional training data.

| Method | Add. Data | Pick Place | Meta-World | | |
|---|---|---|---|---|---|
| | | | Easy | Hard | All |
| DAML [60] | No | 1 | 4 | 8 | 6 |
| T-OSVI [10] | No | 10 | 50 | 7 | 28.5 |
| AWDA [5] | No | **100** | 74 | 11 | 42.5 |
| AWDA [5] | Yes | **100** | 73 | 30 | 51.5 |
| **OSVI-WM (Ours)** | No | **100** | **96** | **71.5** | **83.8** |

Table 2: Real-World experiments: Success rates and execution breakdowns (in %) are reported. T-OSVI* denotes T-OSVI [10] aided with end-effector depth sensing for improved grasping.

| Method | Two-Franka-PP | | | Human-Franka-PP | | | Human-Franka-Push | |
|---|---|---|---|---|---|---|---|---|
| | Overall Success | Breakdown | | Overall Success | Breakdown | | Overall Success | Breakdown |
| | | Reach Obj. | Grasp Obj. | | Reach Obj. | Grasp Obj. | | Reach Obj. |
| T-OSVI [10] | 0 | 52 | 0 | 0 | 0 | 0 | 16 | 96 |
| T-OSVI* [10] | 4 | 52 | 36 | 0 | 0 | 0 | - | - |
| AWDA [5] | 88 | 88 | 88 | **92** | 92 | 92 | 76 | 100 |
| **OSVI-WM (Ours)** | **96** | 96 | 96 | **92** | 92 | 92 | **100** | 100 |

(e) **Human-Franka-Push** (Fig. 5c): This features two colored boxes on a tabletop. The task involves selecting a box and pushing it either forward or backward (four tasks total). We train on three and test on the held-out one. The expert is a human and agent is a Franka arm.

In the pick-and-place tasks above, sequences from test object-target pairs are excluded from training. While this may appear to be an extension of training tasks, most prior methods require training on all pairs to succeed. Generalizing to an unseen pair requires the model to truly follow the expert, rather than rely solely on scene semantics. Additional environmental details are in the appendix.

**Baselines**: As noted in Sec. 1 and Sec. 2, most one-shot visual imitation methods are evaluated on tasks that are minor variations of training tasks. AWDA [5] is the only method that uses the same sensors and environment settings as ours and evaluates on unseen test tasks, making it our primary baseline. We also compare against two widely used methods: DAML [60] and T-OSVI [10]. In real-world experiments, T-OSVI fails to achieve successful grasps. To mitigate this, we augment it with the same depth-based gripper correction used in OSVI-WM and AWDA, where an end-effector-mounted depth camera corrects residual pose errors. We refer to this variant as T-OSVI* and report its results in Table 2. For OSVI-WM, re-planning is used only on the MetaWorld 'hard' tasks, as we observe strong performance even without it on all other benchmarks.

## 4.2 Results

**How well does OSVI-WM generalize to novel, unseen tasks?** We compare OSVI-WM against baseline methods in Table 1 on the Pick-and-Place [10] and Meta-World [61] benchmarks. OSVI-WM is evaluated over 100 rollouts per task with varied object configurations, and success rates are reported. Baseline results are reported from [5]. OSVI-WM outperforms prior methods, achieving success rates up to 30% higher than AWDA even when AWDA uses additional training data. OSVI-WM's ability to predict future latent states enables effective planning, resulting in strong performance even on the Meta-World 'hard' tasks. Failure cases in this setting often occurred in edge cases where objects, such as windows or doors, were positioned far from the agent, making manipulation challenging due to depth ambiguity in external RGB observations. Nonetheless, OSVI-WM achieves a 71.5% success rate on these tasks, outperforming all baselines.

**Can OSVI-WM be used for real-world robotic tasks?** We evaluate OSVI-WM and baselines on the real-world setups, each tested over 25 rollouts using the same 25 object configurations across methods for fair comparison (Table 2). These setups introduce added complexity: in Two-Franka-PP, the expert and agent robots are mounted at different locations, while in Human-Franka-PP and Human-Franka-Push, the expert is a human. Despite these challenges, OSVI-WM achieves consistently high success rates, showing strong real-world applicability. The table also breaks performance into sub-tasks like

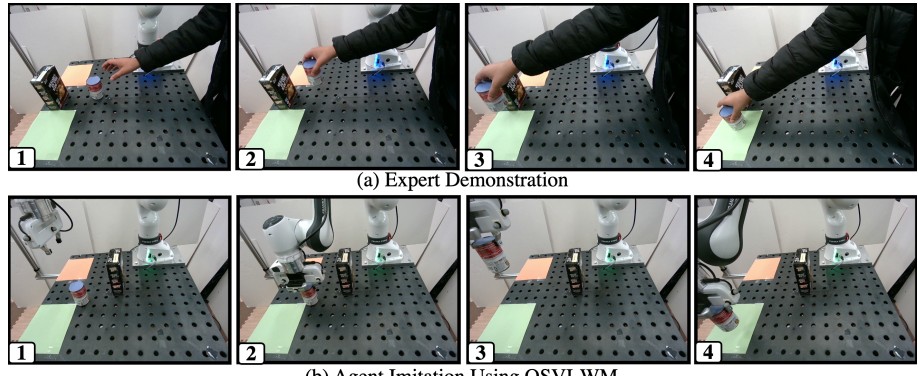

(a) Expert Demonstration

(b) Agent Imitation Using OSVI-WM

Figure 6: **Real-World Qualitative Example**: The human expert demonstrates the test task of picking and placing a can in the green target, which the agent successfully imitates using OSVI-WM.

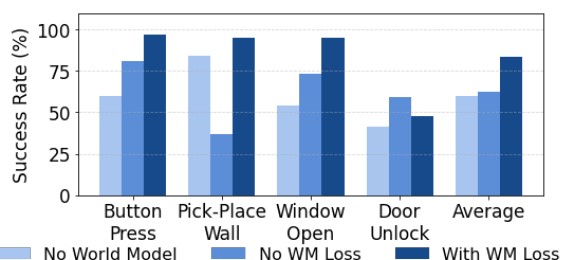

Figure 7: Ablation study on the world model (WM) and WM loss. Using both achieves the best performance.

Figure 8: Ablation Study on OSVI-WM's Re-Planning (RP).

reaching and grasping. In Two-Franka-PP, T-OSVI [10] reaches the object in 52% of trials but fails to grasp it. T-OSVI* (i.e., T-OSVI with grasp correction) improves grasping to 36% but completes the full task only once (4%). AWDA succeeds in 88% of the trials, failing due to incorrect object selection. OSVI-WM performs best, succeeding in 96% of the trials. Human-Franka-PP is more challenging due to the embodiment gap. In this setting, T-OSVI and T-OSVI* fail to reach the correct object, while AWDA and OSVI-WM both succeed in 92 % of the trials. In Human-Franka-Push, all methods consistently reach the correct object. However, due to the low margin for error in this setting, AWDA and T-OSVI often collided with the object during fine manipulation, leading to reduced success rates of 76% and 16%, respectively. OSVI-WM executed these movements more accurately, achieving 100% success. Since there is no grasping involved in this task, we do not run T-OSVI*.

We present a qualitative example of OSVI-WM in the Human-Franka-PP real-world setup. In Fig. 6a, a human expert demonstrates the task of placing a can on a green target. Using only this demonstration, the agent (white Franka arm) successfully completes the task (Fig. 6b), despite changes in object locations and embodiment differences. This task was excluded from training, illustrating OSVI-WM's ability to generalize to unseen tasks by following the expert's intent rather than relying solely on visual semantics. This example showcases OSVI-WM's effectiveness under all key conditions outlined in our problem setup: unseen tasks, embodiment mismatch, unaligned trajectories, and in-domain training. More visualizations on different tasks are in the appendix.

**What is the impact of the world-model-guided trajectory generation module?** We assess the impact of this module through two ablations on the Meta-World dataset. The first removes the world model loss $\mathcal{L}_{wm}$; the second removes the world-model-guided trajectory generation module entirely, replacing it with a single action model that predicts an action vector from the expert demonstration and the agent's initial observation. This vector is passed through the pooling and waypoint prediction modules. As shown in Fig. 7, removing the world model yields the lowest performance, while adding it without $\mathcal{L}_{wm}$ provides limited gains. The loss $\mathcal{L}_{wm}$ is applied to future latent states predicted by the world model and is independent of ground-truth waypoint supervision. It acts as an auxiliary dynamics-consistency regularizer and encourages robust, task-agnostic representations, thus improving generalization to unseen tasks. Hence, combining the world model with $\mathcal{L}_{wm}$ achieves the best performance, underscoring the importance of both the module and its training signal.

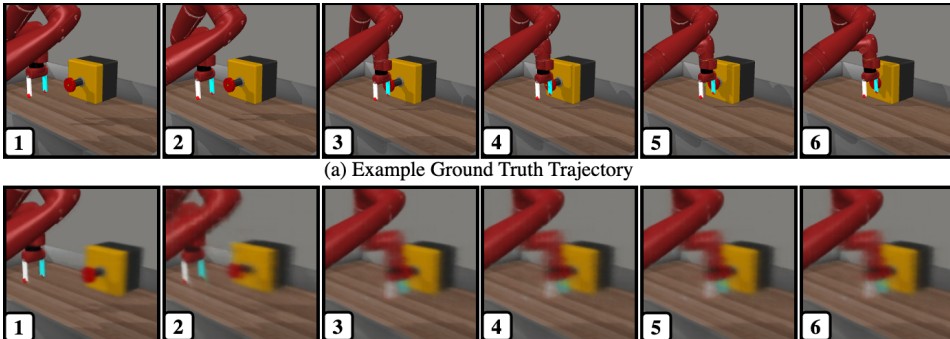

(a) Example Ground Truth Trajectory

(b) OSVI-WM Generated Latent Trajectory Decoded to Images

Figure 9: (a) An example ground truth rollout of the Button-Press-V2 task from Meta-World [61]. (b) Latent trajectory generated using OSVI-WM's world model decoded into images for visualization.

In Fig.9, we visualize the predicted trajectory generated by OSVI-WM's world model on the Button-Press-V2 task [61]. To convert the latent states into images, we train a Transpose Convolution-based image predictor. Images from the task are encoded using OSVI-WM's trained encoder and spatial pooling layer, and the image predictor is trained to predict back the pixels. During evaluation, we generate the future latent states $(r_2, \ldots, r_M)$ using the expert demonstration and initial agent frame, and decode them into images (Fig.9b) using the image predictor. A ground truth rollout is shown for comparison in Fig. 9a. Two key observations emerge: (a) the predicted trajectory captures meaningful, task-completing states, effectively summarizing the task; (b) the latent states are informative enough to reconstruct interpretable images. As expected, later states appear more pixelated due to accumulated errors in recursive latent state predictions. Despite this, the decoded trajectory clearly reflects the full task execution and produces accurate downstream performance.

**Which components of OSVI-WM are most critical to its success?** As shown above, the world-model-guided trajectory generation module, along with supervision from the WM loss $\mathcal{L}_{wm}$, is critical to OSVI-WM's success. Re-planning further boosts performance, particularly on the challenging Meta-World 'hard' tasks. Fig. 8 isolates the impact of re-planning, showing an approximate 30-point gain. For comparison, we also report the average success rate of AWDA [5] (with additional training data). Notably, even without re-planning, OSVI-WM outperforms AWDA. Table 3 presents further ablation studies evaluating key architectural choices. Removing stop-gradient in the WM loss significantly degrades performance, dropping Meta-World accuracy by 7% and Pick-and-Place by 80%, clearly indicating model collapse.

Applying spatial pooling before the world-model-guided trajectory generation block introduces an information bottleneck, resulting in a 85% drop in Pick-and-Place and an 4% drop in Meta-World. Switching from multi-step recursive to single-step trajectory prediction has no effect on Pick-and-Place but lowers Meta-World performance by over 15% due to Meta-World's greater complexity. Together, these results

Table 3: Ablation results (avg. success rates) on the Pick-Place [10] and Meta-World [61], evaluating the impact of different architectural choices by individually disabling them.

| Stop Grad in WM Loss | Spatial Pool after WM | Multi-Step Prediction | Benchmarks | |
|:---:|:---:|:---:|:---:|:---:|
| | | | Pick-Place | Meta-World |
| × | ✓ | ✓ | 20 | 76.5 |
| ✓ | × | ✓ | 15 | 79.5 |
| ✓ | ✓ | × | **100** | 67.3 |
| ✓ | ✓ | ✓ | **100** | **83.8** |

show that OSVI-WM's effectiveness stems from key components: (a) the world-model-guided trajectory generation with WM loss, (b) re-planning, (c) stop-gradient in WM supervision, (d) spatial pooling after the world model, and (e) multi-step trajectory prediction. Regularization strategies like asymmetric demonstration image mixup [5] further improve generalization.

**Division of train and test tasks:** The choice of train–test task division affects performance, especially in the Meta-World simulation environment. During training, the model learns two key components: inferring tasks from the expert and acquiring the skills needed to complete these tasks. Hence, careful consideration is required when choosing training tasks to ensure meaningful generalization to unseen tasks. To further investigate this, we conducted two ablation studies on the Meta-World benchmark. In the first, we excluded all button-related tasks (Button-Press-Topdown, Button-Press-Topdown-Wall, Button-Press-Wall) from training and evaluated on Button-Press. The model still achieved a 97%

success rate, nearly identical to when the button-related tasks were included. This is because button pressing is relatively kinematically less complicated and the required skills were already acquired from other training tasks. In the second ablation, we removed Window-Close from the training set and evaluated on Window-Open. Here, performance decreased from 95% to 80%.The sliding window tasks here involve more complex physical dynamics. So the absence of related training data led to a noticeable drop in generalization. If the training set included a very large variety of tasks with diverse motions and skills, the model would potentially perform well on any task, even if no similar tasks were present in the training set. However, this approach would require large out-of-context datasets, significant training resources, and extended training times. In OSVI-WM, we follow the train–test split proposed in AWDA [5] for consistency with prior work.

## 5   Conclusion

We introduced OSVI-WM, a novel framework for one-shot visual imitation on unseen tasks. Given a single expert demonstration and the agent's initial observation, OSVI-WM encodes them into latent states and uses a world-model-guided approach to predict a trajectory of future latent states. This trajectory is decoded into physical waypoints that control the robot. Extensive experiments in both simulation and real-world settings demonstrate OSVI-WM's strong performance, and ablation studies confirm the importance of key design choices in the framework. By generalizing effectively from a single demonstration to unseen tasks, OSVI-WM takes a step toward enabling autonomous robotic systems capable of operating in complex, real-world domains such as industrial automation and medical assistance, without constant human supervision.

**Limitations**: While OSVI-WM demonstrates promising results, there are limitations that open avenues for future work. Like prior work [10, 5, 34], it only predicts 3D positions and a gripper state, without modeling the end-effector's orientation. This may limit its applicability to tasks requiring rotational actions, such as screwing or side-grasping. The current formulation focuses on single-task execution; extending the framework to sequential or multi-task settings is an interesting direction for future work. The division of train and test task can potentially influence the performance of OSVI-WM as detailed in Section 4. Further, although the test tasks are unseen during training, they belong to the same family of manipulators as the training data. While these limitations do not hinder current performance, they highlight promising directions for enhancing the framework in more complex task settings.

## 6   Acknowledgements

This paper is supported in part by the Army Research Office under grant number W911NF21-1-0155, by the New York University Abu Dhabi (NYUAD) Center for Artificial Intelligence and Robotics, funded by Tamkeen under the NYUAD Research Institute Award CG010, and by NSF under grant number 2208189.

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
