# OpenReview forum: "OSVI-WM: One-Shot Visual Imitation for Unseen Tasks using World-Model-Guided Trajectory Generation"
_NeurIPS.cc/2025/Conference — NeurIPS 2025 poster_

### Official Review · Reviewer_9Vzo · 2025-06-24

**Clarity:** 4
**Significance:** 2
**Originality:** 2
**Rating:** 4
**Confidence:** 4

**Summary:**

This paper proposes a one-shot visual imitation learning method for unseen robotic manipulation tasks. By taking a single expert demonstration as context information, this model generates a sequence of waypoints to complete novel tasks. To enhance the framework's ability for modeling dynamic environments, this paper introduces a world-model-guided trajectory generation proxy loss. The effectiveness of this method is validated through extensive experiments in both simulation and the real world.

**Questions:**

1. Why does the world model loss bring benefits to the one-shot imitation learning on novel scenarios? For example, the Human-Franka-PP task only contains 3 training manipulation tasks; the world model would overfit to these tasks, so why would it have better generalization ability for novel tasks?

2. For dynamic world modeling, various methods exist to reconstruct subsequent frames. Could you provide a comparison with other techniques, like VQ-VAEs and diffusion models, to further assess how well the representation captures dynamic scenes?

3. How do different types of similarity—such as semantic similarity and visual similarity—between the training dataset and validation tasks affect learning outcomes? Furthermore, the impact of various related tasks within the training dataset on overall performance remains underexplored.

**Ethical Concerns:**

["NO or VERY MINOR ethics concerns only"]

**Final Justification:**

I think this paper could be accepted due to the impressive one-shot learning performance.

**Limitations:**

yes

**Paper Formatting Concerns:**

No paper formatting concerns

**Quality:**

3

**Strengths And Weaknesses:**

## Strengths

1. This paper provides sufficient real-world experiments to prove the effectiveness.
2. This paper is well-organized, and the proposed method is clearly explained.
3. It is impressive to imitate human videos for a novel manipulation task.

## Weaknesses

1. Although the experimental results indicate that incorporating the world model loss is beneficial, it remains unclear why this leads to improvements on unseen tasks. Training a model on a small dataset is prone to overfitting, which would be expected to hinder its generalization capabilities on novel tasks. More detailed qualitative and quantitative analyses of the world model loss should be provided.

2. The similarity between training scenarios and novel validation scenarios would significantly influence the performance of one-shot imitation learning systems. However, this lacks a comprehensive analysis of this relationship.

---

> ### Author Rebuttal · Authors · 2025-07-31
>
> We thank the reviewer for the valuable comments. We are happy that you found our human-franka experiments impressive. We address each of the concerns (weaknesses) and questions in the following.
>
> **Concern 1: Although the experimental results indicate that incorporating the world model loss is beneficial, it remains unclear why this leads to improvements on unseen tasks. Training a model on a small dataset is prone to overfitting, which would be expected to hinder its generalization capabilities on novel tasks. More detailed qualitative and quantitative analyses of the world model loss should be provided.**
>
> **Response**: Training a model on a small dataset can indeed lead to overfitting if not properly regularized. Most existing one-shot visual imitation methods are trained end-to-end to predict actions, often relying heavily on image semantics while underutilizing the expert demonstrations. These methods typically emphasize features crucial for action prediction alone. While regularization techniques like asymmetric demonstration image mixup, as used in AWDA [5], have shown improvements in generalization by disrupting this correlation, we enhance this further by incorporating the world model loss. This loss $\mathcal{L}_{wm}$ on the future latent states predicted by the world model is independent of the ground truth actions (actions being waypoints in our case). This additional supervision helps the model focus on learning robust representations of environment dynamics, which improves generalization to unseen tasks.
>
> **Concern 2: The similarity between training scenarios and novel validation scenarios would significantly influence the performance of one-shot imitation learning systems. However, this lacks a comprehensive analysis of this relationship.**
>
> **Response**: Thank you for the suggestion. We agree that the similarity between training scenarios and novel validation scenarios can impact the performance of one-shot imitation learning systems. During training, the model learns two key components: inferring tasks from the expert and acquiring the skills needed to complete these tasks. Generally, the higher the similarity between the skills required in the training tasks and those in the test tasks, the better the generalization. To explore this further, we conducted two new ablation studies:
> * Ablation 1: Button Press Tasks Removed from Training. We removed all training tasks related to button pressing (Button-Press-Topdown, Button-Press-Topdown-Wall, Button-Press-Wall) and evaluated the model on the Button-Press task. The model still achieved a high success rate of 97%, nearly identical to when the button-related tasks were included. This is because button pressing is relatively kinematically less complicated and the required skills were already acquired from other training tasks.
> * Ablation 2: Window Tasks Removed from Training. We removed the Window-Close task from training and evaluated on Window-Open. In this case, performance dropped from 95% to 80%. The sliding window tasks here involve more complex physical dynamics. So the absence of related training data led to a noticeable drop in generalization.
>
> We will include these results and a corresponding discussion in the updated manuscript to better clarify this relationship.
>
> **Question 1: Why does the world model loss bring benefits to the one-shot imitation learning on novel scenarios? For example, the Human-Franka-PP task only contains 3 training manipulation tasks; the world model would overfit to these tasks, so why would it have better generalization ability for novel tasks?**
>
> **Response**: As detailed in the response to your Concern 1, the world model loss provides an additional regularizing training signal, encouraging the model to focus on general features necessary for predicting the next states of the environment, alongside predicting the waypoints. This impact is more evident on larger datasets like Meta-World (as shown in Fig. 7 of the manuscript) compared to smaller datasets like Human-Franka-PP. Increasing the amount of data would certainly help mitigate overfitting. Nonetheless, OSVI-WM achieves high success rates on the novel tasks within Human-Franka-PP, Human-Franka-Push, and Two-Franka-PP (Table 2 of the manuscript), outperforming competing methods.
>
> **Question 2: For dynamic world modeling, various methods exist to reconstruct subsequent frames. Could you provide a comparison with other techniques, like VQ-VAEs and diffusion models, to further assess how well the representation captures dynamic scenes?**
>
> **Response**: VQ-VAE and diffusion models have proven effective for modeling dynamic scenes and have been successfully extended to world models (e.g., [A, B]). Pixel-space world models such as those based on VQ-VAE and diffusion techniques are often computationally intensive to train and deploy. Given the scope and constraints of this rebuttal, we provide a qualitative comparison with these methods based on insights from existing literature. We compare OSVI-WM’s world model against VQ-VAE and diffusion-based world models along several key dimensions:
> 1. World model representation: As discussed in Section 2 of the manuscript, world models can represent the environment’s state either in pixel space or in a latent embedding space. VQ-VAE and diffusion-based world models usually operate in the pixel space, where future frames are generated or reconstructed directly as images. In contrast, OSVI-WM represents the world in a latent embedding space, optimized specifically for downstream prediction tasks such as waypoint planning. Several recent works ([2, 23, 22, 29, 61]) have shown that latent-space representations are more effective for learning meaningful abstractions, as they avoid modeling low-level pixel details that are often irrelevant for downstream tasks.
> 2. Dynamic world: World models are generally designed to capture how environments evolve in response to agent actions. However, they differ in how they handle other dynamic elements (e.g., moving obstacles, background agents). Diffusion models can implicitly model such non-agent-dependent dynamics due to their probabilistic nature, though this comes with significant computational cost. OSVI-WM, in its current form, does not explicitly model dynamics unrelated to the agent. However, it can be extended to do so by introducing an additional environment-dependent learnable latent token, enabling the representation of external scene changes.
> 3. Computational efficiency: OSVI-WM is more efficient, as it bypasses costly pixel-level reconstruction. VQ-VAE and especially diffusion models are, typically, more resource-intensive due to decoder use and iterative generation steps.
> 4. Planning Efficiency: OSVI-WM’s latent embeddings are directly usable for downstream tasks like planning and control. In fact, some recent works (e.g., [61,C,D]) have shown that learned latent states can be effectively combined with model-based controllers without any further training. In contrast, VQ-VAE and diffusion outputs often require an additional encoder to derive planning-compatible representations, adding overhead.
> 5. Interpretibility and visualization: VQ-VAE and diffusion models offer more intuitive visualization since their outputs are in pixel space. OSVI-WM requires a separate decoder to visualize latent states, making interpretation less direct.
> 6. State-Action space: VQ-VAE uses a discrete latent space, while OSVI-WM and diffusion models operate in continuous spaces making them more suitable for continuous control.
>
> We will include a brief discussion of the above in the revised manuscript.
>
> **Question 3: How do different types of similarity—such as semantic similarity and visual similarity—between the training dataset and validation tasks affect learning outcomes? Furthermore, the impact of various related tasks within the training dataset on overall performance remains underexplored.**
>
> **Response**: We conducted new ablation studies (as detailed in the response to your Concern 2) showing that skill similarity, rather than purely visual or semantic similarity, has a stronger impact on generalization. When core skills needed for the test task are present in the training set, performance remains high even if the tasks differ visually or semantically. We'll include these insights and results in the updated manuscript.
>
> **References**
>
> [A] Robine, Jan, Tobias Uelwer, and Stefan Harmeling. "Smaller world models for reinforcement learning." Neural Processing Letters 55, no. 8 (2023): 11397-11427.
>
> [B] Ding, Zihan, Amy Zhang, Yuandong Tian, and Qinqing Zheng. "Diffusion world model: Future modeling beyond step-by-step rollout for offline reinforcement learning." arXiv preprint arXiv:2402.03570 (2024).
>
> [C] Bar, Amir, Gaoyue Zhou, Danny Tran, Trevor Darrell, and Yann LeCun. "Navigation world models." In Proceedings of the Computer Vision and Pattern Recognition Conference, pp. 15791-15801. 2025.
>
> [D] Bai, Yutong, Danny Tran, Amir Bar, Yann LeCun, Trevor Darrell, and Jitendra Malik. "Whole-Body Conditioned Egocentric Video Prediction." arXiv preprint arXiv:2506.21552 (2025).

---

> > ### Comment · Reviewer_9Vzo · 2025-08-06
> >
> > Thanks for the rebuttal. Most of my concerns have been addressed. I maintain the original positive recommendation.

---

> > > ### Author Response · Authors · 2025-08-09
> > >
> > > Thank you very much for your time and consideration.

---

### Official Review · Reviewer_5LPE · 2025-07-02

**Clarity:** 3
**Significance:** 3
**Originality:** 3
**Rating:** 4
**Confidence:** 4

**Summary:**

This paper presents OSVI-WM, a novel one-shot visual imitation learning framework based on world-model-guided trajectory generation, designed to address the limited generalization capability of existing methods on unseen tasks. OSVI-WM integrates expert demonstration videos with the agent's initial observation to predict sequences of latent states and actions through a learned world model, which are subsequently decoded into physical waypoints for task execution. Extensive experiments conducted across both simulated and real-world robotic platforms demonstrate the effectiveness of the proposed method, with results showing that OSVI-WM significantly outperforms existing approaches, particularly on "hard" tasks that exhibit semantic or structural differences. Furthermore, the framework incorporates waypoint re-planning to enhance execution robustness.
The key contributions of this work include: (1) an end-to-end architecture requiring no large-scale pretraining; (2) a dedicated world-model-guided trajectory generation module for one-shot visual imitation; (3) improved test-time robustness through re-planning; and (4) comprehensive experimental validation across diverse environments.

**Questions:**

1.Generalization to Unseen Positions and Objects: The paper states that OSVI-WM can generalize to new tasks with different object positions and across different embodiments, but it does not provide a detailed explanation of how the model achieves this capability. For example, how does the model ensure the accuracy of action prediction when encountering object placements or new object categories that were not seen during training? Additionally, compared to baseline methods, does the robustness of OSVI-WM to changes in object positions stem from the dynamic modeling of the world model? Further clarification is needed.
2.Visualization and Analysis of Latent States and Actions: The paper demonstrates the visualization of latent states through image reconstruction (Figure 9), but lacks an in-depth analysis of the visualization of latent actions and their relationship with states. For example, do latent actions correspond to the movements of the robotic arm and changes in the interacted objects? How do changes in latent states drive action generation? It is suggested to supplement the visualization and analyze the correlation between the two to enhance interpretability.
3.The Significance and Efficiency of Replanning: Replanning is described as “generating new waypoints when the task is not completed,” which seems equivalent to repeated execution until success. If so, this strategy may reduce efficiency in practical applications due to multiple attempts. The triggering conditions for replanning (such as timeouts or deviation thresholds) and its comparison with the success rate of single planning need to be explained. For example, Figure 8 shows that replanning improves performance, but it does not quantify its time cost. If replanning is merely a repeated call to the same module, then its significance is limited.
4.Redundancy in Latent States: Although latent states are designed to be “compact” representations, the features extracted by the Res-Net encoder may contain a large amount of background information unrelated to the task (such as texture or lighting). How can the world model focus on task-relevant features? Moreover, if there is redundancy in the latent space, will it reduce the efficiency of trajectory generation?

**Ethical Concerns:**

["NO or VERY MINOR ethics concerns only"]

**Final Justification:**

I would like to thank the authors for their response. After consideration of the clarifications provided and the additional analyses presented in the response, I maintain my original assessment and scoring of this manuscript.

**Limitations:**

Yes

**Paper Formatting Concerns:**

I did not notice major formatting issues

**Quality:**

3

**Strengths And Weaknesses:**

Strengths：
1.Novel Integration of World Models: OSVI-WM is the first to introduce a world model into one-shot visual imitation learning, establishing an explicit environment dynamics model. By predicting latent states and generating trajectories, it significantly enhances the model’s ability to reason about future states.
2.Superior Generalization Performance: Experimental results demonstrate that OSVI-WM outperforms existing methods on unseen tasks in both simulation and real-world robotic platforms, achieving over 30% improvement in success rates on challenging tasks with semantic or structural variations.
3.Strong Cross-Embodiment Adaptability: The method performs robustly across different embodiments (e.g., human-to-robot imitation) without requiring additional training data or large-scale pretraining, lowering the barrier for real-world deployment.
4.Enhanced Robustness via Replanning: The waypoint re-planning mechanism dynamically adjusts the execution strategy upon failure, further improving task completion rates.


Weaknesses：
1.Lack of End-Effector Orientation Control: The framework's limitation to predicting only 3D positions and gripper states restricts its applicability in tasks that require precise end-effector orientation control. Many real-world manipulation tasks, such as screwing, drilling, or side-grasping, necessitate accurate orientation control.
2.Limited Handling of Dynamic Environments: Although the world model aims to capture environment dynamics, its ability to handle highly dynamic or uncertain environments remains unexplored. In scenarios with multiple moving objects, sudden changes in the environment, or significant variations in object properties, the model's predictions may become less reliable,.
3.Redundancy in Latent States: Although latent states are designed to be “compact” representations, the features extracted by the Res-Net encoder may contain a large amount of redundant information unrelated to the task (such as texture or lighting). This redundant information can interfere with the model's learning and prediction of key task features, reducing the model's efficiency and accuracy. Therefore, further research is needed on how to remove this redundant information from the latent states to improve the model's performance and generalization ability.
4.Limited Theoretical Analysis: The paper lacks a rigorous theoretical analysis of the framework's properties.

---

> ### Author Rebuttal · Authors · 2025-07-31
>
> We thank the reviewer for the valuable comments. Some of the weaknesses listed, such as Limited Handling of Dynamic Environments, Redundancy in Latent States, and Limited Theoretical Analysis, are indeed active research areas within world models. OSVI-WM beats all previous results on one-shot visual imitation. Additionally, the modularity of our framework will facilitate integration of future advancements in the above three research areas. Below, we address each of the concerns (weaknesses) and questions in detail.
>
> **Concern 1: Lack of End-Effector Orientation Control.**
>
> **Response**: As noted in the limitations paragraph in Section 5 of the manuscript, OSVI-WM currently predicts only 3D positions and gripper states, aligning with existing works like [10,5,32]. While our current tasks do not demand precise orientation control, the framework is designed to be flexible. It will be extended in future work to include orientation control by incorporating end-effector orientation data into the waypoints during training. This adaptability ensures that OSVI-WM can be tailored to more complex tasks like screwing, drilling, or side-grasping.
>
> **Concern 2: Limited Handling of Dynamic Environments**
>
> **Response**: The trained world model is generally good at handling object movements directly influenced by the agent's actions, such as a button pressed by a robot (Fig. 9). Even in cases where external agents move objects important to the task, the world model can usually dynamically adapt to such changes appropriately. For other dynamic elements that do not directly affect the agent, like moving people or other robots, probabilistic modeling or optimization over environment-dependent latent variables can be employed. However, there is no widely accepted method for modeling such dynamic movements, and it is an active area of research in world models. Existing works in robotic control [32, 5, 10, 57, 61, 8] typically focus on static environments. While our current approach aligns with this trend, future work could explore integrating more sophisticated models to better handle dynamic and uncertain environments.
>
> **Concern 3: Redundancy in Latent States**
>
> **Response**: As the reviewer correctly notes, latent states are intended to provide compact representations of the environment that focus on the most relevant features. To reduce the influence of irrelevant factors such as texture or lighting, we train the world model directly in the latent embedding space rather than in the image pixel space. Specifically, the loss function (Section 3.3) is applied on the latent state embeddings instead of the decoded image pixels, where cues like texture and lighting tend to dominate. This approach is consistent with recent advances in self-supervised learning [2] and robotics [61, 8], where training in latent space has been shown to better capture task-relevant features while avoiding overemphasis on pixel-level patterns like brightness or texture.
>
> While quantifying redundancy in latent representations remains an active area of research, we argue that for the purposes of the current work, some degree of redundancy in a larger embedding space is not detrimental to performance. This is because the latent embeddings are subsequently passed through pooling and MLP-based waypoint predictor layers, where the MLP learns to attend to task-relevant features. The primary requirement for the latent representation is to effectively disentangle image features. As long as the latent space captures the underlying structure of the scene, the downstream MLP can leverage this disentanglement to generate accurate waypoint predictions. Most learning-based methods, including competing approaches like [5, 32, 10, 57, 60], utilize encoders such as ResNet to extract features. Although redundancy is possible in all of them, it is not widely recognized as a major factor in reducing performance.
>
> **Concern 4: Limited Theoretical Analysis**
>
> **Response**: We acknowledge the importance of theoretical analysis in understanding the properties of any framework. However, providing a rigorous theoretical analysis for large learning-based methods, such as OSVI-WM, is inherently challenging due to their complex and data-driven nature. Consequently, similar to other works in this area, we have focused on empirical validation to demonstrate the effectiveness and robustness of the various design choices in our framework (Section 4.2).
>
> **Question 1: Generalization to Unseen Positions and Objects**
>
> **Response**: As outlined in the ablation studies in Section 4.2 of the manuscript, OSVI-WM’s generalization capabilities arise from several key components: (a) world-model-guided trajectory generation with WM loss, (b) re-planning, (c) stop-gradient in WM supervision, (d) spatial pooling after the world model, and (e) multi-step trajectory prediction. Many one-shot visual imitation methods struggle with generalization because the model plans actions based on the objects in the image while underutilizing the expert demonstration. To discourage this, we employ the world model to predict the next latent states alongside the predicted waypoints. The WM loss applied to these predicted latent states is independent of the waypoints, providing a regularizing effect. Additionally, the use of asymmetric demonstration mixup enhances generalization. We will incorporate these clarifications into the updated manuscript for further clarification.
>
> **Question 2: Visualization and Analysis of Latent States and Actions**
>
> **Response**: The latent actions in our model do indeed correspond to the movements of the robotic arm. We qualitatively analyze this relationship through the following experiments:
> * Effect of Latent Actions on Latent States: We observe that when a zero vector is used as the latent action, the predicted latent state remains unchanged, indicating no movement. In contrast, when a random latent action vector is applied, the visualized latent state exhibits small, unstructured movements of the robotic arm. This qualitatively confirms that latent actions have a direct and controllable influence on latent states.
> * Effect of Latent States on Action Generation: To assess how changes in latent states drive the generation of latent actions, we perform an intervention during multi-step prediction in Eq. 1. Specifically, we manually reset a future predicted latent state $r_j$​ (where $1<j<M−1$) to be equal to the initial state $r_1$​, and feed this into the action encoder. We find that the latent actions adapt accordingly, attempting to recover the trajectory and guide the latent state back toward the originally planned sequence. This demonstrates that the action generation mechanism is sensitive to changes in the latent state and aims to maintain trajectory consistency.
>
> While we are unable to include these new visualizations in the rebuttal due to submission guidelines, we will incorporate them into the final version of the paper to enhance interpretability and provide clearer insights into the interaction between latent states and actions.
>
> **Question 3: The Significance and Efficiency of Replanning**
>
> **Response**: When task execution fails (detected by the simulation environment configuration), OSVI-WM triggers re-planning by taking a new observation and generating a fresh sequence of waypoints based on the robot's current state. The replanning mechanism in OSVI-WM is, therefore, not equivalent to simple repeated execution. Instead of open-loop retries, replanning is a closed-loop process that takes a new observation of the current robot state and generates fresh waypoints based on that feedback. This closed-loop replanning does incur additional time cost. For the results shown in Figure 8, the time comparison is as follows:
> * Without re-planning: Each episode runs for a maximum of 500 steps (~40 seconds on our system with one RTX A4000 GPU, 128 GB RAM, and an Intel i9 CPU).
> * With re-planning: New waypoints are generated every 500 steps, for up to 2000 steps (~160 seconds) total.
>
> Despite this overhead, replanning substantially improves success rates. Notably, in our paper, replanning is only applied for the ‘hard’ Meta-World tasks. Even without replanning, OSVI-WM achieves a success rate of ~42% on these tasks (71.5% after re-planning), which is already higher than all existing baselines (Fig. 8). Additionally, it is important to highlight that many baseline methods, such as [10], are also inherently closed-loop but still perform worse than OSVI-WM, further emphasizing the strength of our approach beyond just repeated planning. We will include a discussion on the time cost of replanning in the revised manuscript.
>
> **Question 4: Redundancy in Latent States**
>
> **Response**: As discussed in our response to your Concern 3, we explicitly address the issue of unrelated background information, such as texture or lighting, by training the world model in the latent embedding space instead of the image pixel space. This design choice ensures that the model focuses on higher-level, task-relevant representations, rather than low-level visual features. Recent works in self-supervised learning and robotics [2, 8, 61] have similarly adopted this approach to improve generalization and reduce sensitivity to irrelevant visual cues.
>
> Similar to the response to your Concern 3, we claim that, while some redundancy in latent features may exist, the primary requirement for the latent representation is to effectively disentangle image features. As long as the latent space captures the underlying structure of the scene, the downstream MLP can leverage this disentanglement to generate accurate waypoint predictions. Most learning-based methods, including competing approaches like [5, 32, 10, 57, 60], utilize encoders such as ResNets to extract features. Although redundancy is possible in all of them, it is not widely recognized as a major factor in reducing performance.

---

> > ### Comment · Reviewer_5LPE · 2025-08-06
> > **Thanks for your response**
> >
> > Thank you for your detailed and prompt response! You have answer my concerns implicitly. I will maintain my score

---

> > > ### Author Response · Authors · 2025-08-09
> > >
> > > Thank you very much for your time and consideration.

---

### Official Review · Reviewer_DoHi · 2025-07-03

**Clarity:** 3
**Significance:** 3
**Originality:** 3
**Rating:** 4
**Confidence:** 3

**Summary:**

This paper introduces OSVI-WM, a novel framework for one-shot visual imitation learning (OSVI) designed to generalize to unseen tasks. The core problem it addresses is that existing OSVI methods often fail when test tasks are semantically or structurally different from training tasks. Also, the agent and the expert have different embodiments and there is no direct alignment between their states or actions. The proposed solution is a model that, given a single expert demonstration video and the robot's initial observation, uses a learned action model (to infer the intention of the task shown by the expert and output a latent action) and a learned world model to plan a sequence of future latent agent states corresponding to the task. The sequence is then decoded into a series of physical waypoints for the robot's controller to imitate and execute. The entire framework, including the world model, is trained end-to-end on in-domain data without requiring massive external video pretraining. The authors demonstrate through experiments on simulated and real-world robotic platforms that OSVI-WM significantly outperforms prior methods, particularly on challenging, unseen tasks.

**Questions:**

1. The action model is an inverse dynamic model (IDM) conditioned on given tasks, and the world model is a corresponding forward dynamics model (FDM), in the context of latent action learning [1-3]. Could you elaborate on the benefit of this architecture, versus a single, larger model that directly predicts the next latent states $r_{i+1}$ from the history and the expert demonstration? Other design choices also include, directly learning a seq2seq-style connector, to translate the expert latent state sequence into corresponding agent space [4, 5]. Does the factorization of action model and dynamics model encourage learning a better representation or extracting the task semantics than other approaches?
2. The world model here is conditioned on the expert videos (tasks), so it is more like a goal-conditioned future state predictor rather than pure dynamics model. While effective, I think the framing could be more precise.
3. The process of transforming the generated waypoints into low-level robot actions requires further clarification (Attributed Waypoint Prediction in Section 3.2). For example, how does the waypoint controller computes low-level control commands for execution using inverse kinematics of the robot?
4. The current method assumes the availability of aligned image-state pairs in the agent sequence. Could the authors discuss how the proposed approach could be adapted to a setting where only image-action trajectories are provided, where proprioceptive states of the agent are replaced by robot actions?

[1] Ye W, Zhang Y, Abbeel P, et al. Become a proficient player with limited data through watching pure videos[C]//The Eleventh International Conference on Learning Representations. 2022.\
[2] Schmidt D, Jiang M. Learning to act without actions[J]. arXiv preprint arXiv:2312.10812, 2023.\
[3] Bruce J, Dennis M D, Edwards A, et al. Genie: Generative interactive environments[C]//Forty-first International Conference on Machine Learning. 2024.\
[4] Mazzaglia P, Verbelen T, Dhoedt B, et al. GenRL: Multimodal-foundation world models for generalization in embodied agents[J]. Advances in neural information processing systems, 2024, 37: 27529-27555.\
[5] Wang Y, Yu R, Wan S, et al. FOUNDER: Grounding Foundation Models in World Models for Open-Ended Embodied Decision Making[C]//Forty-second International Conference on Machine Learning. 2025.

**Ethical Concerns:**

["NO or VERY MINOR ethics concerns only"]

**Final Justification:**

I would like to thank the authors for their detailed response. I maintain my positive score.

**Limitations:**

The authors should be more precise about the boundaries of their method's generalization capabilities. Acknowledging that the "unseen" tasks are still "in-domain" (i.e., from the same family of manipulations as the training data) would provide a more nuanced and accurate picture of the current work's scope.

**Quality:**

4

**Strengths And Weaknesses:**

Strengths:
1. The paper tackles a crucial and difficult problem in robotics: one-shot generalization to new tasks from a single video. Moving beyond simple variations in object positions to tasks that are semantically distinct is a significant step towards more capable and flexible robots.
2. The technical quality of the paper is high. The architecture is well-designed and the experimental evaluation is exceptionally thorough, including challenging human-to-robot imitation with embodiment mismatch. The comprehensive ablation studies are also convincing.
3. The core idea lies in learning a meta-action-model to infer the semantics of the given task and output high-level latent actions, and integrating a world model to generate future latent trajectory plans given the actions and the expert video, which is then decoded it into low-level waypoints. This is a well-motivated approach.

Weaknesses:
1. The re-planning mechanism boosts performance on hard tasks, but it need manual assessment in real-world autonomy, requiring a human in the loop. An automated failure detection mechanism is needed for re-planning in real-world application.
2. The paper's main claim is generalization to unseen tasks, but the "unseen" test tasks in are semantic or combinatorial variations of tasks seen during training. This is a valid and challenging setup, but it leaves open the question of how the model would perform on tasks that are truly out-of-domain (e.g., training on pick-place tasks and testing on a wiping or drawer-opening task). The generalization claim, while supported by the experiments, may be limited to new tasks within the same family of manipulations.

---

> ### Author Rebuttal · Authors · 2025-07-31
>
> We thank the reviewer for the detailed review, acknowledging the strengths of the paper, and for the constructive comments. We address each of the concerns (weaknesses) and questions in the following.
>
> **Concern 1: The re-planning … application.**
>
> **Response**: OSVI-WM uses re-planning only for Meta-World 'hard' tasks (Section 4.1). Since these are simulation tasks, success is automatically verified by the environment configuration. Although we do not apply re-planning in real-world experiments, enabling autonomous re-planning would require automated task success detection. Some prior works have explored vision-language models [A,B] or reward/value networks [22] for this purpose. We will include a discussion on this in the updated manuscript. Notably, even without re-planning, OSVI-WM surpasses the performance of existing methods (Fig. 8).
>
> **Concern 2: The paper's … manipulations.**
>
> **Response**: During training, the model essentially learns two components: to infer tasks from the expert and to acquire the skills necessary to complete these tasks. In completely out-of-domain tasks, the model might still grasp the task context from the expert but may not learn the required skills if the training and testing tasks involve completely different skill sets. For instance, pick-and-place tasks involve skills like reaching, grasping, moving, and placing objects, whereas wiping requires maintaining contact with a surface while moving the end-effector. This skill disparity can lead to failures in testing if the model is trained only on pick-and-place tasks. To further support this, we conducted two additional ablations:
> * Ablation 1: We removed all training tasks related to button pressing (Button-Press-Topdown, Button-Press-Topdown-Wall, Button-Press-Wall) and evaluated the model on the Button-Press task. The model still achieved a high success rate of 97%, nearly identical to when the button-related tasks were included. This is because button pressing is relatively kinematically less complicated and the required skills were already acquired from other training tasks.
> * Ablation 2: We removed the Window-Close task from training and evaluated on Window-Open. In this case, performance dropped from 95% to 80%. The sliding window tasks here involve more complex physical dynamics. So the absence of related training data led to a noticeable drop in generalization.
>
> However, if the training set included a very large variety of tasks with diverse motions and skills, the model would potentially perform well on any task, even if no similar tasks were present in the training set. Achieving this would require large datasets, high training resources, and training times. We will add a discussion on this in the updated manuscript to clarify how the generalization scales to different unseen tasks.
>
> **Question 1-A: The action model is an inverse dynamic model (IDM) conditioned on given tasks, and the world model is a corresponding forward dynamics model (FDM), in the context of latent action learning [1-3].**
>
> **Response**: The world model functions like a forward dynamics model, while the action model resembles an inverse dynamics model, with the distinction that OSVI-WM’s action model is conditioned on expert videos rather than the agent’s future states ($a_i = \pi(r_i, \dots, r_1, e_N, \dots, e_1)$).
>
> **Question 1-B: Could you elaborate … approaches?**
>
> **Response**: Directly predicting the next latent state using only ground truth actions can cause the model to overfit to those actions, resulting in the model only learning features necessary for such action prediction. This dependency often results in poorer generalization to unseen test tasks (Fig. 7). In contrast, incorporating a world model loss provides auxiliary signals that help the model learn environment dynamics independently of any tasks. While a seq2seq-style connector directly learns the ground truth states, the factorization between the action and dynamics (world) model offers key advantages. The dynamics model depends only on the latent actions and current agent states, not directly on the goals, thus enhancing the model's interpretability and modularity and encouraging better representation learning. Additionally, if the goal representation changes (e.g., to text), only the actions model needs retraining, leaving the world model intact.
>
> **Question 2: The world model … precise**
>
> **Response**: OSVI-WM’s world model is conditioned solely on the latent actions and the agent’s states, rather than directly on the expert videos ($r_{i+1} = \mathcal{W}(r_i, a_i, \dots, r_1, a_1)$). The action model, however, is conditioned on the expert videos ($a_i = \pi(r_i, \dots, r_1, e_N, \dots, e_1)$). By making this separation, we get key advantages as detailed in the response to your Question 1-B. We will make this difference clear in the updated manuscript.
>
> **Question 3: The process of … robot?**
>
> **Response**: The generated waypoints represent the end-effector’s 3D positions in the camera’s coordinate frame and binary gripper states (Section 3.2). These positions are transformed into the robot frame using a known camera-to-robot transformation. We denote these 3D positions in the robot’s frame as $p_k = (x_k, y_k, z_k)$, where $k = \{1, 2, \ldots, q\}$. The desired rotation $\mathrm{rot}_k$ of the end-effector (which remains constant in our case) is concatenated to $p_k$ to obtain 6D desired pose $H_k = (x_k, y_k, z_k, \mathrm{rot}_k)$. We use this desired pose with the controller as detailed below:
> * Simulation Experiments: Obtaining the joint angles $\theta_k$ involves solving the inverse kinematics problem , $\mathrm{F.K.}(\theta_k) = H_k$. Given that the robots used in our experiments have more than 6 degrees of freedom, the inverse kinematics does not have a unique solution. Therefore, we utilize the following optimization problem:
> $$\min_{\theta_k} \left\| H_k - \mathrm{F.K.}(\theta_k) \right\|$$
> where $\theta_k$ is the robot's joint angles and $\mathrm{F.K.}$ is the forward kinematics. To avoid jerky motion from large pose jumps between waypoints, we interpolate sub-waypoints between the current and target waypoint poses. The controller tracks these sub-waypoints, while the gripper follows the predicted gripper state.
>
> * Real-World Experiments: We employ a differential inverse kinematics approach, optimizing for joint velocity by solving:
> $$\min_{\dot{\theta}_k} \left\| J \dot{\theta}_k - K_p e \right\|^2 + \lambda  \left\|(I - J^\dagger J)(\dot{\theta}_k - \dot{\theta}_b) \right\|^2$$
> $J$: Jacobian of the end-effector; $\dot{\theta}_k$: joint velocity; $K_p$: proportionality constant; $e$: pose error between the current and target poses. The second term in the equation above ensures that the joint velocities remain close to a nominal joint velocity $\dot{\theta}_b$ (please refer to [C] for more details). The resulting velocities are passed to the Franka robot, running an internal proprietary controller at approximately 4 kHz. As in simulation, sub-waypoints ensure smooth motion, and the gripper executes the predicted open/close action.
>
>
>
> **Question 4: The current … actions?**
>
> **Response**: In most robotics datasets [58,10,D,E,F,G,H] and real data collected in the lab, access to the robot’s proprioceptive states, typically joint angles and angular velocities, is standard. These states can be converted into end-effector poses and velocities via the robot’s forward kinematics. In OSVI-WM, we utilize these end-effector poses to generate ground-truth waypoints, which serve as target actions for training the model as detailed in the manuscript. However, in the uncommon scenario where proprioceptive states are unavailable, robotic datasets may represent actions as the difference between successive joint angles (or end-effector poses). We can still reconstruct the full joint trajectory by cumulatively summing these deltas starting from a known initial joint configuration (which is feasible with current-day robot hardware, which provides reasonable sensor resolutions; methods to reconstruct estimates of the robot pose from external cameras are also available). Using forward kinematics, we can then convert the reconstructed joint angles at each time step into end-effector poses, which are subsequently used to generate ground-truth waypoints for training OSVI-WM. This reconstruction-based approach allows us to maintain the overall framework and learning objectives of OSVI-WM even in the absence of direct proprioceptive state data.
>
> **Limitations: The authors … scope.**
>
> **Response**: Thank you for this feedback. We will include a discussion of this in the limitation section, along the lines of the response to your concern 2.
>
> **References**
>
> [A] Liu, Haichao, et al. "Robodexvlm: Visual language model-enabled task planning and motion control for dexterous robot manipulation." arXiv preprint arXiv:2503.01616 (2025).
>
> [B] Xiong, Chuyan, et al. "Autonomous interactive correction MLLM for robust robotic manipulation." In 8th Annual Conference on Robot Learning. 2024.
>
> [C] Lynch, Kevin M., and Frank C. Park. Modern robotics. Cambridge University Press, 2017.
>
> [D] Nasiriany, Soroush, et al. "Robocasa: Large-scale simulation of everyday tasks for generalist robots." arXiv preprint arXiv:2406.02523 (2024).
>
> [E] Abhishek Gupta, et al.. “Relay policy learning: Solving long-horizon tasks via imitation and reinforcement learning”. arXiv preprint arXiv:1910.11956, 2019. 2, 5, 17
>
> [F] Vikash Kumar. “Manipulators and Manipulation in high dimensional spaces”. PhD thesis,
> University of Washington, Seattle, 2016.
>
> [G] Kumar, Vikash, et al. "Robohive: A unified framework for robot learning." Advances in Neural Information Processing Systems 36 (2023): 44323-44340.
>
> [H] Khazatsky, Alexander, et al. "Droid: A large-scale in-the-wild robot manipulation dataset." arXiv preprint arXiv:2403.12945 (2024).

---

> > ### Author Response · Authors · 2025-08-07
> >
> > With the discussion period coming to a close soon, we wanted to kindly check if you have any remaining questions or concerns regarding our work. We would be happy to provide any additional clarifications or answer questions that you might have.

---

> > ### Comment · Reviewer_DoHi · 2025-08-09
> >
> > Thank you for the detailed response. Most of my concerns are addressed. I will maintain the positive score.

---

### Official Review · Reviewer_9img · 2025-07-03

**Clarity:** 2
**Significance:** 2
**Originality:** 2
**Rating:** 4
**Confidence:** 2

**Summary:**

This paper tackles poor generalization in one-shot visual imitation learning by proposing a framework guided by a learned world model. Instead of direct policy learning, the method plans a trajectory of states and actions in the world model's latent space, which is then decoded into physical waypoints for execution. This approach is designed to better handle unseen tasks with different underlying contexts. Evaluated across simulated and three real-world platforms, the method reportedly demonstrates consistent and significant improvements over prior art, with over a 30% performance gain in some scenarios.

**Questions:**

- In the simulated experiments, it seems that how to divide the train and test tasks will dramatically affect the experimental results. For instance, the Button-Press task shares very similar action behaviors with the Button-Press-Topdown task. So what are the considerations behind this split? How are the similar tasks in the trainset different from the testing tasks?
- What are the specific settings of Human-Franka tasks? What is the camera setup? Are the objects fully randomized on the tabletop?

**Ethical Concerns:**

["NO or VERY MINOR ethics concerns only"]

**Limitations:**

yes

**Quality:**

3

**Strengths And Weaknesses:**

## Strengths
- Technically solid method pipeline. One-shot transfer to unseen tasks is a challenging topic.
- Comprehensive experiments conducted on 2 sim and 3 real-world benchmarks.
- Conducting Human-to-Robot skill transfer, where the embodiment gap is quite large.

## Weaknesses
- Lack of comparison against one-shot imitation learning methods through alignment and open-loop replay, e.g., DINOBot.
- I am not familiar with world-model-based methods for manipulation. While Human-to-Robot skill transfer is impressive to me, the studied manipulation tasks are generally simple, i.e., top-down grasping with low precision requirements. Given that BC-based or VLA-based policies have enabled very dexterous manipulation behaviors, I believe a discussion is necessary to illustrate the high-level pros and cons of this world-model-based approach when compared with more prevalent and straightforward BC-based methods.
- Please refer to the questions for more minor weaknesses.

DINOBot: Robot Manipulation via Retrieval and Alignment with Vision Foundation Models. ICRA'24.

---

> ### Author Rebuttal · Authors · 2025-07-31
>
> We thank the reviewer for the detailed review and for the constructive comments. We further appreciate your recognition of the strengths of the paper. In the following, we address each of the concerns (weaknesses) and questions.
>
> **Concern 1: Lack of comparison against one-shot imitation learning methods through alignment and open-loop replay, e.g., DINOBot.**
>
> **Response**: DINOBot (and similar pixel alignment and open-loop replay methods) perform one-shot imitation by using a gripper-mounted camera to visually compare the current scene to demonstration frames. It retrieves the closest visual match and estimates pixel-level alignment to grip the object and execute a trajectory. While its use of foundation models for visual retrieval followed by pixel-level alignment is unique and effective without further training, its problem setup differs from ours. DINOBot relies solely on an ego-centric gripper camera, whereas OSVI-WM uses external third-person cameras. Consequently, DINOBot has shortcomings against external camera based methods as discussed below. Moreover, adapting DINOBot’s retrieval and pixel alignment framework for external cameras is not trivial. Therefore, we restricted our comparisons to one-shot visual imitation methods that use external third-person camera views. The main shortcomings we observed when attempting to use DINOBot are detailed below.
> 1. To conduct retrieval and pixel alignment, DINOBot requires the target object to be visible to the gripper camera at all times. In our experiments, the target is often initially outside the gripper’s field of view, making such assumptions invalid.
> 2. In multi-object settings like our pick-and-place tasks, which involve four randomly arranged objects and four targets, both the current observation and demonstration images contain multiple objects with different configurations. As a result, there is no single, consistent visual transformation that aligns the current scene’s pixels to that of the demonstration. DINOBot’s reliance on clear visual correspondence for a single object breaks down in such cluttered and compositional environments.
>
> We will include this discussion on DINOBot and similar works in the updated manuscript to clarify the differences.
>
> **Concern 2: I am not familiar with world-model-based methods for manipulation. While Human-to-Robot skill transfer is impressive to me, the studied manipulation tasks are generally simple, i.e., top-down grasping with low precision requirements. Given that BC-based or VLA-based policies have enabled very dexterous manipulation behaviors, I believe a discussion is necessary to illustrate the high-level pros and cons of this world-model-based approach when compared with more prevalent and straightforward BC-based methods.**
>
> **Response**: Thank you for your suggestion. We will include a discussion on the pros and cons of our world-model-based approach compared to straightforward BC-based or VLA-based approaches in the updated manuscript. Generally, both world-model-based and BC-based (or VLA-based) approaches ultimately aim to execute tasks by predicting actions. However, the key distinction is that world-model-based approaches also learn the world dynamics independently of the task. This discourages the model from overly fixating on action prediction and encourages learning more generic features through the additional training signal provided by the world model loss. Most existing methods, whether world-model-based or BC-based, typically focus on training and testing within the same set of tasks, as detailed in our manuscript. Many of these methods have demonstrated good results in complex dexterous manipulation tasks, as you mentioned. However, when faced with out-of-distribution tasks, like those in our examples, existing BC-based methods such as [5, 10, 57] often show low success rates, even on top-down grasping tasks. Additionally, our experience indicates that even recent VLA-based robot foundation models like Pi0 [A] and Groot [B] require in-context retraining for new tasks for accurate manipulation, despite being pre-trained on large datasets.
>
> **Question 1-A: In the simulated experiments, it seems that how to divide the train and test tasks will dramatically affect the experimental results. For instance, the Button-Press task shares very similar action behaviors with the Button-Press-Topdown task.**
>
> **Response**: How one divides the train-test task will indeed impact results, particularly in the Meta-World simulation environment. In OSVI-WM, to be consistent with prior work, we adopt the train-test split proposed in AWDA [5] for Meta-World. During training, the model learns two key components: inferring tasks from the expert and acquiring the skills needed to complete these tasks. Hence, careful consideration is required when choosing training tasks to ensure meaningful generalization to unseen tasks. To explore this further, we conducted two new ablation studies:
> * Ablation 1: Button Press Tasks Removed from Training. We removed all training tasks related to button pressing (Button-Press-Topdown, Button-Press-Topdown-Wall, Button-Press-Wall) and evaluated the model on the Button-Press task. The model still achieved a high success rate of 97%, nearly identical to when the button-related tasks were included. This is because button pressing is relatively kinematically less complicated and the required skills were already acquired from other training tasks.
> * Ablation 2: Window Tasks Removed from Training. We removed the Window-Close task from training and evaluated on Window-Open. In this case, performance dropped from 95% to 80%. The sliding window tasks here involve more complex physical dynamics. So the absence of related training data led to a noticeable drop in generalization.
>
> If the training set included a very large variety of tasks with diverse motions and skills, the model would potentially perform well on any task, even if no similar tasks were present in the training set. However, this approach would require large out-of-context datasets, significant training resources, and extended training times.
>
> **Question 1-B: So what are the considerations behind this split?**
>
> **Response**: Based on Ablation 1 and Ablation 2, we could, in principle, construct numerous train-test splits and get different results. However, to maintain consistency with prior work, we adopt the split proposed in AWDA [5] for Meta-World.
>
> **Question 1-C: How are the similar tasks in the trainset different from the testing tasks?**
>
> **Response**: As mentioned in Section 4.1 of the manuscript and detailed further in Section C of the appendix, the similar tasks in the training set differ from those in the testing set in Meta-World either through the introduction or omission of distractors such as walls (for the ‘easy’ tasks), or by requiring a different set of motions (for the ‘hard’ tasks), which can only be executed correctly by following the expert demonstration.
>
> **Question 2: What are the specific settings of Human-Franka tasks? What is the camera setup? Are the objects fully randomized on the tabletop?**
>
> **Response**: In the Human-Franka tasks (Human-Franka-PP and Human-Franka-Push), the Franka agent is positioned on a tabletop, with a camera fixed near the table's edge at a height that provides a clear view of the table and the objects, as depicted in Figs. 5b and 5c. We utilize an Intel RealSense D455 camera with known intrinsic and extrinsic (camera-to-robot transformation) parameters, capturing images at 640x480 resolution. Additionally, an Intel RealSense D435 is mounted on the robot's end-effector to assist with grasping, as detailed in Section 3.2 of the manuscript. Object locations are fully randomized during both training and testing. For testing, to ensure fairness across all methods, we use the same 25 random object configurations across all methods.
>
> **References**
>
> [A] Black, Kevin, Noah Brown, Danny Driess, Adnan Esmail, Michael Equi, Chelsea Finn, Niccolo Fusai et al. "$\pi_0 $: A Vision-Language-Action Flow Model for General Robot Control." arXiv preprint arXiv:2410.24164 (2024).
>
> [B] Bjorck, Johan, Fernando Castañeda, Nikita Cherniadev, Xingye Da, Runyu Ding, Linxi Fan, Yu Fang et al. "Gr00t n1: An open foundation model for generalist humanoid robots." arXiv preprint arXiv:2503.14734 (2025).

---

> > ### Comment · Reviewer_9img · 2025-08-05
> >
> > Thanks for the rebuttal. Most of my concerns have been addressed. I maintain the original positive recommendation.

---

> > > ### Author Response · Authors · 2025-08-09
> > >
> > > Thank you very much for your time and consideration towards our work.

---

### Note · Authors · 2025-08-12

We sincerely thank the reviewers and area chair for their time and effort. In response to the reviewers' feedback, we conducted new experiments and provided detailed point-by-point responses to the raised concerns and questions. We appreciate that the reviewers have acknowledged that the bulk of their concerns have been overall addressed through our responses and additional experiments. We will incorporate these additional experiments and comments into the final version of the manuscript.

---

### Decision · Program_Chairs · 2025-09-17

**Decision:**

Accept (poster)

**Comment:**

This submission presents OSVI-WM, a one-shot imitation framework for robot manipulation tasks. The framework uses a world model (WM) to guide the generation of a trajectory (a sequence of latent states and actions) based on a (human) video demonstration. The trajectory is then decoded into a sequence of waypoints that the robot tracks during execution. The approach claims to exhibit higher generalization from train to test task distributions.

All reviewers were initially weakly on the accept side and had questions about the details of the method, the experiment setup, train and test task distributions, and the mechanism by which generalization arises in the proposed method. In response, the authors have allayed the reviewers' concerns by providing additional explanations and, importantly, doing additional valuable ablation studies. The reviewers didn't have any major specific questions after the rebuttal but also didn't increase their "weak accept" scores.

The metareviewer thinks that from the standpoint of solving robot manipulation the proposed approach falls somewhat short: it essentially does motion planning but doesn't exhibit advanced object manipulation, currently lacking precise gripper control. However, this weakness is offset by this submission's well-executed and well-analyzed use of WMs to enable generalization. While the approach does need additional work to be competitive in manipulating objects, the robot manipulation community will benefit from the insights provided by OSVI-WM regarding WM use in this domain.